# TAR: TOKEN ADAPTIVE ROUTING FRAMEWORK FOR LLMS TOKEN-LEVEL SEMANTIC CORRECTION INSPIRED BY NEURO-LINGUISTIC PATHWAYS

## ABSTRACT

Large language models (LLMs) often suffer from cascading errors in math reasoning due to token-level semantic defects. A key limitation is that the reliance on unidirectional feedforward pathways makes LLMs unable to dynamically correct token-level defects during reasoning. In contrast, neuro-linguistic pathways in the human brain—centered on Broca's and Wernicke's areas—operate as a closed loop, integrating semantics through feedforward pathways while leveraging feedback circuit for error correction and signal adaptation. The loop involves conflict detection in the anterior cingulate cortex (ACC), cross-regional error transmission via the arcuate fasciculus/IFOF, and compensatory reprocessing in the DLPFC–Broca circuit. Inspired by the functional architecture of neuro-linguistic pathways, we propose a Token Adaptive Routing (TAR) framework that establishes a brain-inspired self-correcting loop in LLMs without requiring parameter fine-tuning. TAR comprises three components: (1) **Semantic Defect Monitor**, analogous to the anterior cingulate cortex (ACC) for identifying tokens with semantic defects; (2) **Adaptive Router**, resembling the arcuate fasciculus/IFOF for routing defective tokens to the most compatible LLM functional block; and (3) Feedback-based Re-representation, inspired by the DLPFC–Broca circuit for correcting semantic defects. Experiments show that TAR improves accuracy and reduces the number of inference tokens. On the challenging AIME25 benchmark, TAR improves the accuracy of Qwen3-1.7B by +3.36% while reducing inference tokens by 13.7%. Furthermore, we reveal that maintaining high token confidence is essential for reasoning performance, and deeper blocks in LLMs play a crucial role in shortening reasoning depth. Our code is available at here

## 1 INTRODUCTION

Human cognitive abilities critically rely on closed-loop coordination between feedforward pathways and feedback circuits across distributed brain regions (Markov et al., 2014; Alitto & Usrey, 2003; Shen et al., 2022; Furutachi et al., 2024), as illustrated in Figure 1(a). Feedforward pathways progressively extract and integrate perceptual information, whereas feedback circuits return cross-hierarchical signals and correct errors, enabling efficient representations and flexible adaptation to external stimuli (O'Reilly et al., 2021; Hockley et al., 2025; Perez et al., 2025). Recent studies show that artificial neural networks—especially large language models (LLMs)—exhibit striking parallels with cortical feedforward processing in their hierarchical representations and context modeling (Li et al., 2023b; Caucheteux et al., 2023; Kumar et al., 2024; Lei et al., 2025). However, unlike biological neural systems, current LLMs generally lack feedback-like self-correcting loops, which limits dynamic information integration and error correction and makes them prone to biases in complex long-sequence reasoning.

A primary reason for LLMs inference error lies in token-level semantic defects. As illustrated in Figure 1(b), even when crucial cues are present in the preceding context, they may fail to be properly integrated into the representation of the relevant token; the resulting erroneous representation then propagates through subsequent computation and triggers cascading reasoning errors (Biran et al., 2024; Ye et al., 2024b). Existing mitigation strategies typically introduce self-correction mechanisms or increase the depth of deliberation (e.g., chain-of-thought, multi-step reflection) (Ding

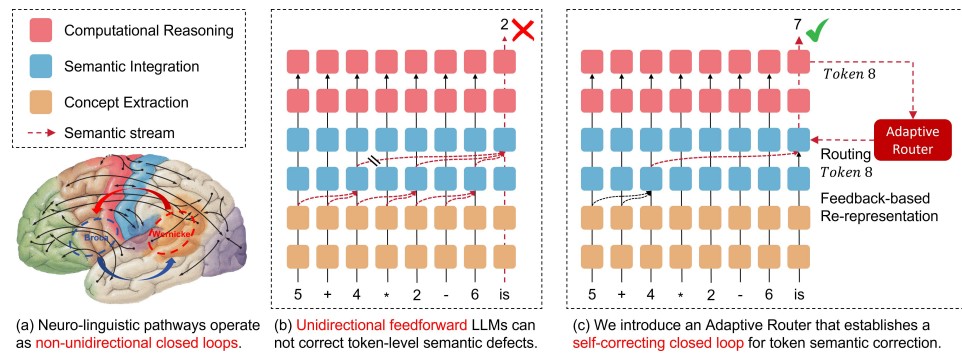

(a) Neuro-linguistic pathways operate as non-unidirectional closed loops.

(b) Unidirectional feedforward LLMs can not correct token-level semantic defects.

(c) We introduce an Adaptive Router that establishes a self-correcting closed loop for token semantic correction.

Figure 1: Inspired by neuro-linguistic pathways, we introduce a token router into LLM that leverages its native semantic representation ability to enable self-repair of semantic defects.

et al., 2024; Ye et al., 2024a; Guo et al., 2025). However, these approaches often incur substantial computational and training overheads. Consequently, under limited resources, effectively alleviating token-level semantic defects becomes a central challenge for improving the robustness of reasoning.

Prior studies have shown that errors caused by token-level semantic defects can often be detected in advance through internal signals, even before the model generates its final outputs (Li et al., 2023a; Ye et al., 2024a; Fu et al., 2025). For example, LLMs can leverage differentiated patterns in their internal representations to "sense" potential mistakes and, to some extent, exhibit internal features akin to "regret" (Ye et al., 2024a). DeepConf (Fu et al., 2025) measures reasoning confidence by analyzing the distribution of token logits and discards low-confidence trajectories. Although such methods demonstrate the possibility of anticipating inference biases in LLMs, the absence of feedback loops still prevents timely correction of already deviated token-level semantics.

Existing studies have attempted to construct rule-based feedback loops in LLMs by sending deep-layer token signals back to earlier layers to improve reasoning performance. For instance, in the Two-Hop Queries task, Hopping Too Late (Biran et al., 2024) routes reasoning tokens that have already propagated to deeper layers back to shallow layers and performs another forward pass, which significantly improves reasoning accuracy. This result provides initial empirical evidence that feedback-loop mechanisms can be effective in mitigating token-level semantic defects. However, this approach heavily relies on manual design and intervention, indiscriminately forwarding deep tokens to shallow layers. As a result, it lacks efficiency and generality, making it difficult to scale to more complex long-sequence tasks or larger models.

To design a more efficient and general self-correcting loop for LLMs that mitigates token-level semantic defects, we draw inspiration from the structure of the brain's neuro-linguistic pathways. At the circuit level, language-related cortical regions centered on Broca's and Wernicke's areas operate as closed loops of feedforward and feedback connections (Nasios et al., 2019; Crosson, 2021), enabling robust and flexible semantic processing. As illustrated in Figure 2, **the anterior cingulate cortex (ACC)** in the prefrontal cortex is responsible for conflict detection, identifying potential errors and activating feedback signals (Carter & Van Veen, 2007). These erroneous signals are then transmitted through long-range tracts such as the **arcuate fasciculus or inferior fronto-occipital fasciculus (IFOF)** to relevant functional regions (Almairac et al., 2015; Houston et al., 2019). Finally, the **dorsolateral prefrontal cortex (DLPFC) and Broca's area** form a compensatory circuit that re-represents and corrects the erroneous signals (Boschin et al., 2017; Hertrich et al., 2021). Together, the **ACC–IFOF–DLPFC/Broca loop** constitutes a self-correcting semantic circuit that provides the neural basis for efficient and reliable language comprehension and production. By analogy, as shown in Figure 1(c), **we introduce a similar self-correcting loop into LLMs, allowing tokens with semantic defects to be selectively routed back to the most compatible representational regions of LLMs**. This enables token-level semantic correction and alleviates cascading errors triggered by semantic defects.

Specifically, guided by the above neuro-linguistic mechanisms, we propose the Token-Adaptive Routing (TAR) framework, which consists of three components: (1) **Semantic Defect Monitor**, analogous to the ACC, that identifies **which** tokens exhibit potential semantic defects; (2) **Adap-**

**tive Router**, analogous to the arcuate fasciculus/IFOF, that determines **where** (which block) these tokens should be routed to for semantic correction; (3) **Feedback-based Re-representation**, corresponding to the compensatory role of the DLPFC–Broca circuit, specifies **how** defective tokens are re-represented and corrected. These three components establish a brain-inspired self-correcting semantic mechanism within LLMs and form a "which–where–how" functional loop.

We summarize the contributions of our work: 1. **Modeling the LLM Routing Problem**: We empirically validate the learnability of token routing, propose an "ability–requirement" vector formulation, and establish confidence as the core signal for both defect detection and routing effectiveness. 2. **LLM Self-Correcting Loop**: We propose the Token Adaptive Routing (TAR) framework, which constructs a closed loop of detection–routing–re-representation within LLMs—without fine-tuning backbone parameters—to correct semantic defects and enhance performance on complex tasks autonomously. 3. **Stable Router Training Strategy**: We design a loss-feedback–driven training strategy that alleviates the instability and sparse feedback issues of the router, which improves the stability and generalization of routing policy learning. 4. **Mechanistic Insights for Token Routing**: Through biased random routing experiments, we reveal that maintaining high token confidence is essential for reasoning performance, and deeper blocks in LLMs play a crucial role in shortening reasoning depth. These findings provide fresh empirical evidence for understanding the functional contributions of internal representations in LLM reasoning.

## 2 PRELIMINARIES

This section characterizes token-level semantic defects and introduces token confidence as a quantitative indicator for detecting them. We then demonstrate the learnability of token routing in Appendix D and formalize it by jointly modeling LLM blocks' representational abilities and defective tokens' semantic requirements All of these establish the theoretical foundation for our Token Adaptive Routing (TAR) framework.

### 2.1 TOKEN-LEVEL SEMANTIC DEFECTS AND CONFIDENCE

During complex reasoning, LLMs often produce token-level semantic defects arising from limited attention allocation (Yu et al., 2025) and unstable semantic decision-making (Wang et al., 2025). Such defects typically occur where semantic aggregation is incomplete or reasoning uncertainty is high. For example, in multi-hop reasoning (Biran et al., 2024), semantic information from one hop may fail to integrate into the next hop's token, resulting in incomplete representation that triggers cascading errors.

Prior studies show that tokens with semantic defects can be identified through internal signals before final output generation (Li et al., 2023a; Ye et al., 2024a; Fu et al., 2025). In particular, the sharpness of a token's predictive distribution reflects the model's certainty at the current reasoning step, with **token confidence** (Fu et al., 2025) serving as a key quantitative indicator of this property. Our empirical analysis reveals a strong negative correlation between token confidence and cross-entropy (CE) loss during supervised fine-tuning. As illustrated in Figure 4 in Appendix B, for tokens $T_i$ at positions $i \in (100, 1000)$, CE loss approaches zero when confidence exceeds 20, but may become larger when confidence falls below 10. This suggests that token confidence directly reflects alignment with ground-truth labels, providing an effective proxy for semantic correctness. So we argue that low-confidence tokens embody potential semantic defects—their incomplete or inconsistent representations fail to produce sharp probability peaks, making them prone to error propagation.

We define token confidence as follows. For a token $T_{i,L}$ at position $i$ produced by the final block $L$, its predictive distribution is denoted by $P_i$. The confidence of this token is then defined as:

$$C_i = \frac{1}{k} \sum_{j=1}^{k} P_i^{(j)}, \tag{1}$$

where $P_i^{(j)}$ denotes the $j$-th largest probability in $P_i$. Higher confidence indicates sharper peaks and more reliable representations; lower confidence corresponds to flatter distributions and more potential semantic defects.

## 2.2 PROBLEM FORMULATION OF TOKEN ROUTING

The core problem of Token Routing is to re-send defective tokens (low-confidence tokens) to the most compatible LLM blocks for semantic correction. To this end, we jointly model blocks' representational abilities and defective tokens' requirements to identify optimal routing targets.

**Ability vector of LLM blocks** Similar to neuro-linguistic pathways in the human brain (Gazzaniga, 2009), LLM blocks are hierarchically organized, each specializing in different representational abilities (Li et al., 2023b; Kumar et al., 2024; Wendler et al., 2024; Lei et al., 2025). Since each block may serve multiple functions, we represent block $l$'s functionality as a $d$-dimensional ability vector $\mathbf{f}_l \in \mathbb{R}^d$, where $f_l^{(j)}$ denotes block $l$'s strength on ability $j$ (e.g., named entity recognition, numerical operations, semantic aggregation).

**Requirement vector of low-confidence tokens** For low-confidence token $T_{i,L}$, we define a $d$-dimensional requirement vector $\mathbf{r}_i \in \mathbb{R}^d$, where $r_i^{(j)}$ reflects the importance of ability $j$' for correcting the token's semantic defect.

**Routing affinity score** To quantify the match between the representational abilities of block $l$ and the semantic requirements of token $T_{i,L}$, we define the routing affinity score as:

$$S(i,l) = \mathbf{r}_i^T \mathbf{f}_l = \sum_{j=1}^{d} r_i^{(j)} \cdot f_l^{(j)}. \tag{2}$$

Accordingly, the optimal routing target block for token $T_{i,L}$ is:

$$l^{*i} = \arg\max_{l \in \{1,\ldots,L\}} S(i,l). \tag{3}$$

This formulation enables precise alignment between tokens' semantic requirements and blocks' representational strengths, providing targeted opportunities for token semantic correction. The following section elaborates on learning both ability and requirement vectors and implementing efficient token adaptive routing.

## 3 METHOD

In this section, we introduce the Token Adaptive Routing (TAR) framework, inspired by human neuro-linguistic pathways, to create a self-correcting loop for addressing token-level semantic defects in LLMs. Then, to mitigate instability and sparse feedback in training the Adaptive Router, we propose a loss-feedback-driven training strategy to enhance the stability and effectiveness of routing policy learning.

### 3.1 BRAIN-LIKE ARCHITECTURE DESIGN OF TOKEN ADAPTIVE ROUTING FRAMEWORK

As outlined in Section 2.1, LLMs often suffer from token-level semantic defects, where incomplete or inconsistent token representations cause cascading errors in reasoning. Existing methods, such as self-reflection or parallel reasoning, mitigate these issues by encouraging deeper reasoning or post-hoc error correction. However, these approaches generate numerous additional tokens, increasing accuracy at the cost of significant computational and training overhead. This motivates an **endogenous self-correction mechanism** that dynamically corrects token-level defects during reasoning, improving efficiency while maintaining strong reasoning performance.

Human language processing benefits from complex neuro-linguistic pathways and functional specialization. Language-related cortical regions, centered on Broca's and Wernicke's areas, form closed loops of feedforward and feedback connections, enabling robust and flexible semantic processing. Feedforward pathways extract and integrate semantic information, while feedback pathways transmit corrective signals to earlier regions, preventing error accumulation. As shown in Figure 2, the anterior cingulate cortex (ACC) detects conflicts and activates feedback signals. These signals are then transmitted through long-range tracts, such as the arcuate fasciculus or inferior fronto-occipital fasciculus (IFOF), to relevant functional regions. Finally, the dorsolateral prefrontal cortex (DLPFC) and Broca's area form a compensatory circuit that re-represents and repairs the

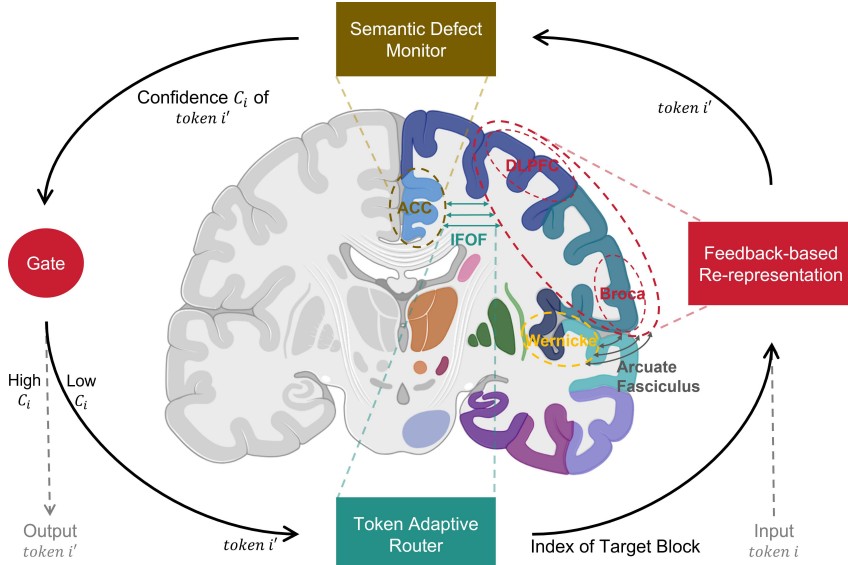

Figure 2: Brain-inspired Token Adaptive Routing (TAR) framework. The design is motivated by neuro-linguistic pathways. The ACC detects conflicts, error signals are routed via the arcuate fasciculus/IFOF, and the DLPFC–Broca circuit performs re-representation and correction.

erroneous signals. Together, the ACC–IFOF–DLPFC/Broca loop forms a self-correcting semantic circuit, enabling efficient and reliable language processing.

While LLMs share hierarchical representation similarities with biological systems (Kumar et al., 2024; Lei et al., 2025), their strictly feedforward architecture lacks feedback loops, limiting their ability to correct token-level semantic defects internally. To overcome this, we propose the **Token Adaptive Routing** (TAR) framework, inspired by neuro-linguistic closed-loop mechanisms, to introduce a brain-like self-correcting loop in LLMs. As depicted in Figure 2, TAR comprises three core components, each inspired by a function of the biological neuro-linguistic pathways: (1) **Semantic Defect Monitor**, analogous to the ACC, that identifies **which** tokens exhibit potential semantic defects; (2) **Adaptive Router**, analogous to the arcuate fasciculus/IFOF, that determines **where** (which block) these tokens should be routed to for semantic correction; (3) **Feedback-based Re-representation**, corresponding to the compensatory role of the DLPFC–Broca circuit, specifies **how** defective tokens are re-represented and corrected. These components create a "which–where–how" loop: detecting defective tokens (which), selecting their correction destination (where), and performing re-representation (how). This establishes a brain-inspired self-correcting mechanism within LLMs.

**Semantic Defect Monitor** This component identifies tokens with potential semantic defects and forwards them to the routing stage for correction. As discussed in Section 2.1, token confidence reflects alignment with ground-truth labels and indicates semantic integrity (Li et al., 2023a; Ye et al., 2024a; Fu et al., 2025). Based on observation (as shown in Figure 4 in Appendix B), the Semantic Defect Monitor detects defects by measuring token confidence: if the confidence of token $T_{i,L}$ falls below a predefined threshold, it is sent to the Adaptive Router; otherwise, if confidence exceeds the threshold or the maximum routing limit is reached, the token is output directly.

**Adaptive Router** This module determines the optimal LLM block for correcting a defective token. Formally, the routing decision for token $T_{i,L}$ is:

$$a_i = \pi_\theta(T_{i,L}; \mathbf{M}; \mathbf{T}), \tag{4}$$

where $\mathbf{M} = [M_1, \ldots, M_L]$ denotes ability messengers from all blocks, $\mathbf{T} = [T_{0,L}, \ldots, T_{i-1,L}]$ represents historical context tokens, and $a_i$ is the index of the selected target block. As shown in Figure 3(a), the router acts as a bypass to the main LLM. When the confidence of token $T_{i,L}$ falls below the threshold, the Adaptive Router computes the target block index $a$ that can best compensate for the semantic defect of $T_{i,L}$, and then routes the token to the block $a$ for subsequent

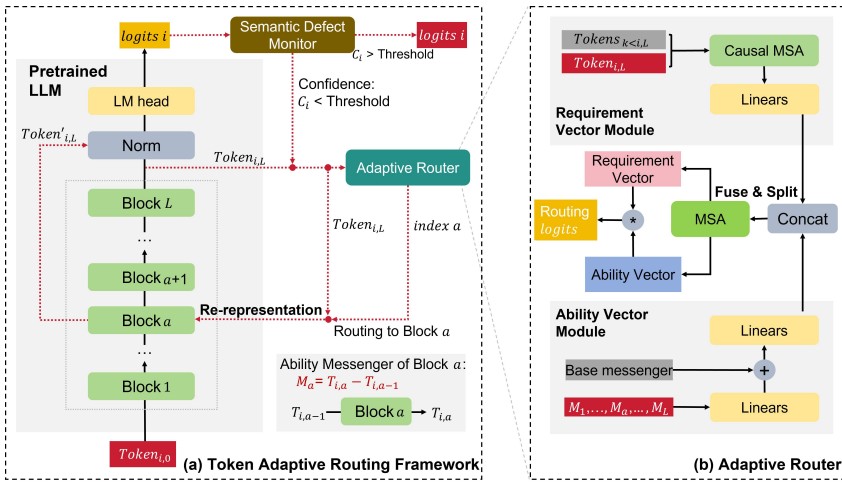

Figure 3: Illustration of the Adaptive Router. (a) Adaptive Router is embedded as a bypass to the LLM for rerouting low-confidence tokens. (b) Internal structure of the requirement and ability vector modules.

re-representation. Inspired by the arcuate fasciculus and IFOF in transmitting error signals to appropriate brain regions (Mesulam, 1990; Friederici, 2011), we design routing based on matching ability requirements and representational abilities as discussed in Section 2.2. Specifically, we model each LLM block with an ability vector $\mathbf{f}_l$ and each defective token with a requirement vector $\mathbf{r}_i$, to identify the most compatible block. Figure 3(b) illustrates the internal design of the router, where $\pi_\theta$ is implemented by two complementary modules: the requirement vector module $R_\theta$ and the ability vector module $A_\theta$. To capture contextual dependence as observed in biological feedback pathways (Kirchgessner et al., 2020; Antunes & Malmierca, 2021), $R_\theta$ takes both the current token $T_{i,L}$ and its historical context tokens $T_{k,L}$ ($k < i$) as input, yielding a $d$-dimensional requirement vector $\mathbf{r}_i = R_\theta(T_{i,L}; T_{k,L}, k < i)$. In parallel, $A_\theta$ characterizes block $j$ by observing the difference (Ability Messenger) $\mathbf{M}_j = T_{i,j} - T_{i,j-1}$ between its input token $T_{i,j-1}$ and output token $T_{i,j}$, and maps it into an ability vector $\mathbf{f}_j = A_\theta(\mathbf{M}_j) \in \mathbb{R}^d$. To enhance interaction between $\mathbf{r}_i$ and $\mathbf{f}_j$, both undergo self-attention for feature fusion, yielding refined representations. Finally, the routing affinity score (Eq. 2 and Eq. 3) selects the most compatible block $a^*$, as the routing destination for token $T_{i,L}$.

**Feedback-based Re-representation** This component corrects semantic defects by re-representing defective tokens. Once the Adaptive Router selects target **block** $a \leq L$, the defective token $T_{i,L}$ is re-injected into block $\mathbf{B}_a(\cdot)$ for feedback-based re-representation:

$$T'_{i,L} = \mathbf{B}_a(T_{i,L}). \tag{5}$$

Specifically, token $T_{i,L}$ interacts with block $a$'s key-value cache via causal attention and passes through its feedforward layer, reintegrating dynamic and static semantic information in block $a$. This process compensates for missing semantic components in $T_{i,L}$, enhancing its internal representation. If the **target block is** $a = L + 1$, it means that the token is not re-injected into any block, and we set $\mathbf{B}_a(\cdot) = \mathbf{I}$, where $\mathbf{I}$ denotes the identity transformation. The corrected token then enters the main LLM backbone, through final normalization $\mathbf{N}(\cdot)$ and output layers $fc(\cdot)$, producing updated logits:

$$z_{i,a} = fr_a(T_{i,L}) = fc(\mathbf{N}(\mathbf{B}_a(T_{i,L}))). \tag{6}$$

where $fr_a(\cdot) = fc(\mathbf{N}(\mathbf{B}_a(\cdot)))$ denotes the re-representation function. Finally, $z_{i,a}$ is either re-monitored by the Semantic Defect Monitor or used as output directly, closing the self-correcting loop.

## 3.2 Training Strategy for the Adaptive Router

This subsection presents the loss-feedback-driven training strategy for the Adaptive Router. We outline its training objective, policy exploration and data collection, and stability regularization to ensure efficient and robust token-level semantic correction.

**Training Objective** The Adaptive Router $\pi_\theta$ aims to route tokens to the optimal block for correcting semantic defects. Its challenge is to quantify a token's semantic defect severity and learn an effective routing policy. As shown in Section 2.1, token-level SFT loss or confidence reflects semantic integrity under the current context. Thus, we use token-level SFT loss to measure defect severity during training. In early attempts, we tried to optimize SFT loss with Gumbel-Softmax gradients, but faced instability and sparse feedback (see Appendix E.2). To address this, we propose a **Loss-Feedback-Driven training strategy: evaluate routing quality by comparing SFT loss before and after routing to guide the router toward better policies**. Formally, for token $T_{i,L}$, the optimal routing action $a^*$ is:

$$a^* = \arg\min_{a \sim \pi_\theta} \mathcal{L}_{\text{SFT}}(z_{i,a}, y_i), \tag{7}$$

where $y_i$ is the ground-truth label, $z_{i,a}$ denotes logits under action $a$, and $\mathcal{L}_{\text{SFT}} = -\log p_i$ with $p_i$ as the predicted probability of the ground-truth label $y_i$. In practice, we supervise the router to approximate this optimal policy $a_i^{(\text{gt})}$ using a cross-entropy objective over routing actions:

$$\mathcal{L}_{\text{router}} = \mathcal{L}_{\text{CE}}\big(\pi_\theta(T_{i,L}, \mathbf{M}, \mathbf{T}), a_i^{(\text{gt})}\big), \tag{8}$$

where $\pi_\theta(T_{i,L}, \mathbf{M}, \mathbf{T})$ is the router's predicted distribution over actions for given the token $T_{i,L}$, context $\mathbf{T}$, and ability messengers $\mathbf{M}$; and $a_i^{(\text{gt})}$ is the best action from exploration (see next subsection). Thus, TAR's training objective is to learn routing policies that reduce SFT loss or increase token confidence, which enhances reasoning accuracy and robustness.

**Routing Policy Exploration and Data Collection** For each token $T_{i,L}$, we use an on-the-fly exploration strategy to evaluate all routing policies and collect training data for the Adaptive Router. In an LLM with $L$ layers, each re-representation offers $L+1$ routing options ($L$ blocks + direct output option). Since the optimal policy is unpredictable, we test all $L+1$ policies, recording their logits and SFT losses:

$$\mathbf{Z} = \{z_{i,1}, \ldots, z_{i,L}, z_{i,L+1}\}, \tag{9}$$

$$\mathbf{L} = \{\mathcal{L}_{i,1}, \ldots, \mathcal{L}_{i,L}, \mathcal{L}_{i,L+1}\}, \quad \mathcal{L}_{i,j} = \mathcal{L}_{\text{CE}}(z_{i,j}, y_i), \tag{10}$$

where $y_i$ denotes the ground-truth label of the token at position $i$. By exploring all $L+1$ candidate policies, we identify the optimal routing policy:

$$a_i^{(\text{gt})} = \arg\min_{a \in \{0, \ldots, L+1\}} \mathcal{L}_{i,a}. \tag{11}$$

Finally, a training sample is collected as:

$$\{T_{i,L}, \mathbf{M}, \mathbf{T}, a_i^{(\text{gt})}\}, \tag{12}$$

where $\mathbf{M}$ denotes the set of Ability Messengers, which capture the input–output differences of each block, and $\mathbf{T}$ denotes the historical context tokens. This exploration ensures stable, high-quality supervision for the Adaptive Router.

**Regularization for Routing Stability** We observe that multiple routing policies can reduce SFT loss, not just a single optimal policy $a^*$. Supervising only with hard label $a^*$ limits generalization and ignores the suboptimal policies' value. To address this issue, we add a KL regularization term on top of the base routing loss $\mathcal{L}_{router}$, which enhances training stability and improves the Router's capacity for policy exploration. We first transform SFT losses $\mathbf{L}$ into a probability distribution via softmax:

$$\text{Prob}_{loss} = \text{softmax}(-\mathbf{L}), \tag{13}$$

where higher probabilities indicate better strategies (lower loss). Meanwhile, the Router outputs a predicted distribution over all candidate policies:

$$\text{Prob}_{router} = \text{softmax}(\pi_\theta(T_{i,L}, \mathbf{M}, \mathbf{T})), \tag{14}$$

where $\pi_\theta$ is the Router's output logits. On this basis, the KL regularization loss is:

$$\mathcal{L}_{kl} = D_{KL}(\text{Prob}_{loss} \,\|\, \text{Prob}_{router}), \tag{15}$$

which constrains the Router's predictions to align the ideal strategy distribution. Therefore, the final training objective is:

$$\mathcal{L} = \mathcal{L}_{router} + \lambda \mathcal{L}_{kl}, \tag{16}$$

with $\lambda = 0.5$ balancing the terms. This approach enhances Router stability and leverages diverse strategies for effective routing.

## 4    EXPERIMENT

This section evaluates the proposed Token Adaptive Routing (TAR) framework. Comparative experiments demonstrate TAR's consistent performance improvements on GSM8K (Cobbe et al., 2021), MATH500 (Hendrycks et al., 2021; Lightman et al., 2023), and AIME25 (Balunović et al., 2025). Ablation studies analyze the effects of routing strategies, confidence filtering, and training approaches. Additional analyses of confidence dynamics and qualitative cases provide insights into TAR's mechanisms.

### 4.1    IMPLEMENTATION DETAILS

**Models**: We tested TAR on Qwen2.5-0.5B (Team, 2024) and Qwen3-1.7B (Yang et al., 2025), chosen for their low memory requirements and deployability. Improving these small-scale LLMs carries broader practical significance, as such models are easier to apply in real-world scenarios. **Training Data**: Training uses the SFT paradigm with Prompt-CoT structures. AIME24 (Balunović et al., 2025), and Olympiad (He et al., 2024) datasets are used, with CoTs generated by the respective model and filtered for correctness, yielding 10K high-quality samples. Each model is trained byt the data generated by itself. Each Adaptive Router is trained for one epoch using our loss-feedback-driven strategy. **Test Data**: Evaluation covers GSM8K, MATH500, and AIME25, spanning varied mathematical reasoning difficulties. **GPUs**: Training and inference use one NVIDIA L40S 48GB GPUs with the Transformers framework. **Evaluation Metrics**: Metrics follow DeepScaleR Luo et al. (2025), with sampling hyperparameters aligned with Qwen series settings.

### 4.2    COMPARATIVE EXPERIMENT

Table 1 shows that the Adaptive Router improves accuracy across GSM8K, MATH500, and AIME25 without altering backbone parameters. In non-thinking mode, TAR boosts accuracy (+1.3% on GSM8K, +3.5% on AIME25). In thinking mode, it achieves higher accuracy with shorter reasoning chains (+1.9% on MATH500 with –4.5% length, +3.2% on AIME25 with –13.6% length), demonstrating that TAR effectively corrects token-level defects and reduces redundant reasoning.

Table 1: Comparison of baseline models and our method (w/ Adaptive Router) on mathematical reasoning benchmarks. We report average Pass@1 and token length in thinking and non-thinking modes. Maximum sequence length is 8192 for GSM8K and MATH500, and 38912 for AIME25.

| Data | Model | Think Mode | Token Length | Pass@1 |
|------|-------|-----------|--------------|--------|
| GSM8K | Qwen2.5-0.5B | Non-thinking | **260** | 41.6% |
|       | **Qwen2.5-0.5B W/ Router (ours)** | Non-thinking | 264 | **42.9%** |
| MATH500 | Qwen3-1.7B | Thinking | 3264 | 82.5% |
|         | **Qwen3-1.7B W/ Router (ours)** | Thinking | **3117** | **84.4%** |
| AIME25 | Qwen3-1.7B | Thinking | 17898 | 36.8% |
|        | **Qwen3-1.7B W/ Router (ours)** | Thinking | **15455** | **40.0%** |
|        | Qwen3-1.7B | Non-thinking | **1878** | 9.8% |
|        | **Qwen3-1.7B W/ Router (ours)** | Non-thinking | 1949 | **13.3%** |

### 4.3    ABLATION EXPERIMENTS

**Routing Strategy** Table 3 and Figure 7 in Appendix E.1 evaluate routing strategies and confidence filtering on AIME25. We compare the Adaptive Router against four random baselines—Right-skewed (deeper blocks), Normal (middle layers), Uniform (all layers), and Left-skewed (shallow blocks)—under three filtering settings: Decreased, No filtering, and Increased. The Adaptive Router outperforms random baselines (40.0% vs. 36.7% best random). Among random strategies, right-skewed routing disrupts reasoning, while Left-skewed maintains accuracy but does not reduce reasoning depth. Interestingly, when Right-skewed routing is combined with the Increased filtering strategy, the model not only recovers near-baseline accuracy but also achieves a shorter reasoning chain. These observations highlight two broader principles: **(i) token confidence provides a reliable signal for evaluating routing effectiveness, and (ii) deeper blocks in LLMs play a crucial**

**role in abstract reasoning and reducing reasoning depth**. For more related work, please refer to Appendix E.1.

**Training Strategy** Table 3 in Appendix E.2 assesses training strategies on Qwen2.5-0.5B with GSM8K. The baseline achieves 41.6% Pass@1. When Gumbel-Softmax is applied to approximate discrete routing decisions, training becomes unstable and feedback signals remain sparse, leading to degraded performance (40.3%). In contrast, Our supervised objectives, $\mathcal{L}_{\text{router}}$ and $\mathcal{L}_{\text{router}} + \lambda \mathcal{L}_{kl}$, improve performance to 42.2% and 42.9%, respectively. This confirms that our loss-feedback-driven strategy stabilizes and enhances routing policy learning. More details can be found in Appendix E.2.

We conducted additional experiments, including analyses of confidence changes (Appendix E.3) and qualitative case studies of reasoning processes (Appendix E.4). For more experiments results, please refer to Appendix E.

## 5 CONCLUSIONS

We propose the Token Adaptive Routing (TAR) framework, a brain-inspired closed loop for LLMs that corrects token-level semantic defects without altering backbone parameters. TAR targets low-confidence tokens using three components: a **Semantic Defect Monitor** identifying defects via confidence signals, an **Adaptive Router** directing tokens to compatible layers via "requirement–ability" matching, and **Feedback Re-representation** reinjecting tokens for corrective processing. This addresses the absence of feedback in feedforward LLMs. Experiments and ablation studies show that: (i) TAR improves performance across GSM8K, MATH500, and AIME25 for various model scales and inference modes, while reducing reasoning length in thinking mode; (ii) The Increased filtering policy—accepting only tokens with improved confidence after routing—is critical, highlighting that **confidence gain is the key signal for effective routing**; (iii) Random routing distributions reveal functional distinctions across layers, with **deep blocks in LLM playing a central role in reducing redundant reasoning**. Overall, TAR provides a biologically inspired, decoupled, and internally self-correcting mechanism for LLMs, offering a new pathway toward more robust and efficient reasoning under limited computational resources.

## 6 RELATED WORK

**Neuro-Linguistic Mechanisms**: Language processing in the human brain relies on interactive closed-loop circuits rather than purely feed-forward flows. Dual-stream models highlight Broca's and Wernicke's areas connected via the arcuate fasciculus (Friederici, 2011; Hickok & Poeppel, 2007). The dorsal pathway supports phonological–motor integration and rapid auditory feedback, while the ventral pathway (IFOF) underpins semantic processing (Duffau et al., 2013; Sarubbo et al., 2013). Through recurrent interactions, these pathways enable cross-verification between phonological form and semantic content. The anterior cingulate cortex (ACC) further contributes conflict monitoring and error-correction signals, recruiting frontal regions when mismatches or violations occur (Shenhav et al., 2013; Abutalebi & Green, 2016). Lots of research shows that the central role of feedback is in self-monitoring and adaptive control during language use. **Token-Level Semantic Detection and Correction**: Reliable detection of token-level semantic correctness is a prerequisite for effective intervention. Internal signals such as log-probabilities, entropy, and loss correlate with factual accuracy and reasoning stability. Higher average token confidence predicts stronger factuality and instruction adherence, while entropy spikes often mark branching errors in reasoning (Mahaut et al., 2024; Fu et al., 2025). Building on such signals, recent work explores token-level interventions during inference. Back-patching analyses demonstrate that reinjecting intermediate token representations from deeper layers to earlier ones can recover failures in multi-hop reasoning (Biran et al., 2024). Other approaches integrate symbolic rules with neural decoding, for example, forcing backtracking in code generation when constraints are violated (Nye et al.). These studies collectively suggest that token-level correction offers a promising pathway for improving robustness in LLM inference.

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

# A USE OF LLMS

In this study, large language models (LLMs) were used exclusively as auxiliary tools for improving the readability of the manuscript. Specifically, LLMs assisted in grammar checking, language polishing, and translation. All ideas, methods, analyses, and conclusions presented in this paper are original contributions of the authors.

# B RELATIONSHIP BETWEEN CONFIDENCE AND CE LOSS

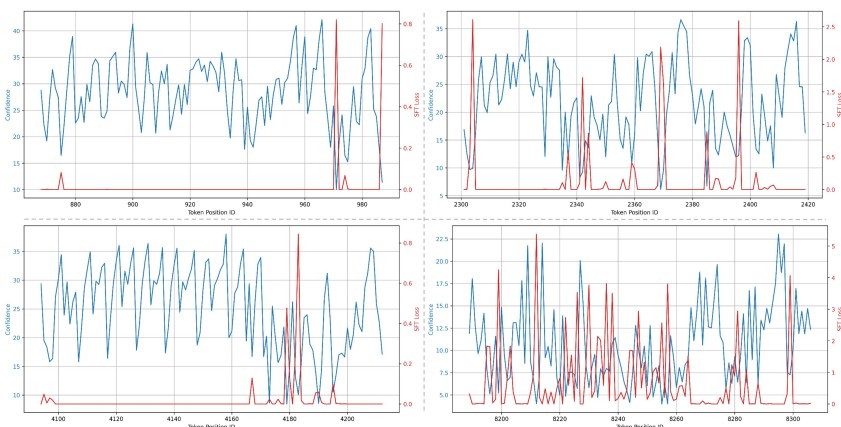

Figure 4: Relationship between token confidence and token-level CE loss (SFT loss). High-confidence tokens (above 20) correspond to near-zero CE loss, while low-confidence tokens (below 10) exhibit substantially higher CE loss.

# C IMPLEMENTATION OF SELF-CORRECTION LOOP

The self-correction loop specifies how TAR operates during inference. When a token $T_{i,L}$ is identified as defective by the Semantic Defect Monitor, it is routed by the Adaptive Router to a target block $a$ for feedback-based re-representation. This process produces updated logits $z_{i,a}$ (Eq. 6). Instead of directly replacing the original logits, the Semantic Defect Monitor first compares the confidence of $z_{i,a}$ with that of the original logits $z_{i,L+1}$. If the confidence has improved (i.e., $C(z_{i,a}) > C(z_{i,L+1})$), then $z_{i,a}$ is adopted as the refined logits; otherwise, the original logits are retained.

The selected logits are then re-examined to determine whether further correction is needed. If the confidence remains below a predefined threshold $\tau$, the token may be routed again to another block for additional re-representation. This iterative rerouting progressively enhances the semantic reliability of low-confidence tokens while avoiding unnecessary updates.

The self-correction loop terminates under two conditions: (*i*) if the confidence of the selected logits exceeds the threshold ($C_i > \tau$), the token is finalized as the output; (*ii*) if the number of rerouting steps reaches a maximum limit $m$, the current logits are used as the output. Together, these rules balance correction accuracy and computational efficiency, ensuring that TAR improves reasoning robustness without introducing excessive overhead.

# D EXISTENCE AND LEARNABILITY OF OPTIMAL TOKEN ROUTING POLICIES

As illustrated in Figure 5, we collect real-time statistics showing how forwarding tokens to different layers affects their semantic reliability. The results demonstrate the *existence of an optimal routing policy*: for each token, there always exists at least one forwarding strategy that either increases token confidence or reduces SFT loss, providing direct evidence for the learnability of routing decisions.

| Token Index | Original Confidence | Optim Routing Policy for confidence | Confidence after Optim Routing | Original SFT loss | Optim Routing Policy for SFT loss | SFT loss after Optim Routing |
|---|---|---|---|---|---|---|
| 4703 | 14.6876 | Block 22 | 15.5438 | 0.0001 | Block 22 | 0.0000 |
| 4704 | 22.6938 | Block 13 | 22.7875 | 0.0000 | Block 0 | 0.0000 |
| 4705 | 16.8063 | Block 1 | 17.7500 | 0.0000 | Block 1 | 0.0000 |
| 4706 | 19.8001 | Block 20 | 20.4719 | 0.0001 | Block 1 | 0.0000 |
| 4707 | 21.2313 | Block 8 | 21.4562 | 0.0000 | Block 25 | 0.0000 |
| 4708 | 18.5625 | Block 22 | 18.6564 | 0.0000 | Block 0 | 0.0000 |
| 4709 | 20.2562 | Block 14 | 20.4937 | 0.0000 | Block 0 | 0.0000 |

Figure 5: Empirical evidence of the existence of optimal routing policies. Each row corresponds to a token, and the results show that forwarding to specific layers can either reduce the SFT loss or increase the token confidence

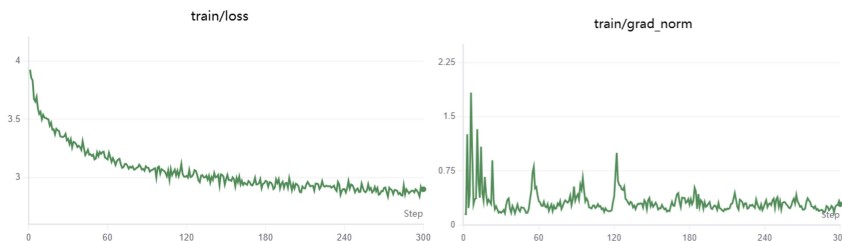

Figure 6: Convergence of router training. Left: training loss decreases steadily, showing effective optimization. Right: gradient norm stabilizes, suggesting improved training stability.

Furthermore, the convergence of the training curves in Figure 6 demonstrates the *learnability* of such policies. Specifically, the training loss steadily decreases and the gradient norm stabilizes over steps, indicating that the Adaptive Router can effectively approximate optimal routing decisions through supervised optimization. Together, these results establish both the existence and the learnability of optimal routing strategies within our framework.

# E  ADDITIONAL EXPERIMENTS

## E.1  ROUTING STRATEGY

Table 3 and Figure 7 systematically analyze the impact of different routing strategies and confidence filtering policies on reasoning performance by AIME25. This analysis serves two purposes: first, to validate the effectiveness of the proposed **Adaptive Router**; and second, to leverage random routing experiments as a probe into the underlying representational mechanisms of LLMs.

We design four types of random routing baselines (Figure 7): **Right-skewed**, which tends to route tokens to deeper blocks; **Normal**, which concentrates routing around middle layers; **Uniform**, which evenly distributes tokens across all layers; and **Left-skewed**, which favors routing to shallower blocks.

On top of these baselines, we further evaluate three confidence filtering policies: (*Decreased*) accepting tokens whose confidence decreases after rerouting; (*No filtering*) allowing all rerouted tokens; and (*Increased*) accepting only tokens whose confidence improves.

This design allows us to examine three central questions: (i) whether adaptive, requirement-driven routing outperforms random strategies; (ii) whether confidence-based filtering is essential for stabilizing and amplifying the benefits of rerouting; and (iii) how different layer regions contribute to routing effectiveness.

Experimental results reveal that random routing produces unstable accuracy and sequence length, while the Adaptive Router consistently achieves higher accuracy with more balanced lengths. In

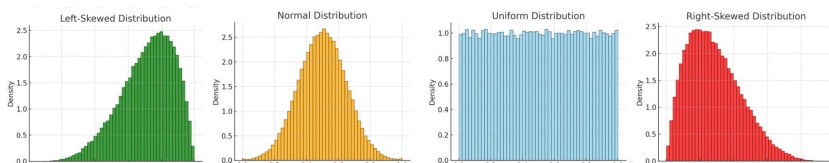

Figure 7: Illustration of four routing distributions used for constructing random baselines. Left-skewed distributions preferentially route tokens to shallow blocks; Normal distributions concentrate around middle layers; Uniform distributions spread evenly across all layers; while Right-skewed distributions bias tokens toward deeper blocks. These distributions are employed to probe the role of different layer regions in token routing.

Table 2: Impact of routing strategies and confidence filtering on AIME25. We report both Pass@1 accuracy (%) and average token length. Random baselines are constructed by sampling routing decisions from different distributions, while the Adaptive Router is our proposed method.

| Routing Strategy | Confidence Filtering | | |
| --- | --- | --- | --- |
| | Decreased | No filtering | Increased |
| Random (Right-skewed) | 0.0% / 1014 | 0.0% / 790 | 36.7% / 9378 |
| Random (Normal) | 6.7% / 2379 | 10.0% / 2336 | 36.7% / 13706 |
| Random (Uniform) | 10.0% / 2601 | 13.3% / 2898 | 36.7% / 14086 |
| Random (Left-skewed) | 33.3% / 14866 | 36.7% / 15657 | 36.7% / 14467 |
| Adaptive Router (ours) | **36.7%** / 15212 | **40.0%** /15857 | **40.0%** / 15455 |

particular, under the *Increased* policy, the performance gains are most significant, indicating that **effective routing must align with confidence improvements** to enhance semantic reliability, rather than simply increasing computational cost.

Interestingly, the four random distributions exhibit distinct patterns: Right-skewed routing almost completely disrupts reasoning ability, while Left-skewed routing has little impact on accuracy but fails to shorten reasoning depth. However, when the *Increased* policy is applied on top of the Right-skewed strategy, the model not only recovers to near-baseline accuracy but also achieves a notably shorter reasoning chain. This phenomenon validates our hypotheses from two perspectives: (i) **token confidence is a critical signal for determining routing effectiveness**; and (ii) **deeper blocks play a key role in abstract reasoning and reducing reasoning depth**.

## E.2 TRAINING STRATEGY

Table 3 reports the results of different training strategies on `Qwen2.5-0.5B` evaluated by `GSM8K`. The base model achieves a Pass@1 of 41.6%. Since routing decisions are discrete and implemented via an *argmax* operation, gradients cannot be directly propagated back to the router. In early versions, we attempted to use the Gumbel–Softmax approximation to make the routing differentiable and propagate the SFT loss end-to-end. However, this approach led to instability and sparse feedback, preventing the router from effectively learning forwarding strategies, and performance dropped to 40.3%.

By contrast, our proposed supervised objective $\mathcal{L}$router alleviates these issues and improves Pass@1 to 42.2%. Adding the KL regularization term, $\mathcal{L}$router$+\lambda\mathcal{L}_{kl}$, further enhances performance, reaching 42.9%. These results demonstrate that aligning the router's policy distribution with the loss-induced distribution not only stabilizes training but also amplifies the benefits of adaptive routing, thereby validating the effectiveness of our *loss-feedback–driven training scheme*.

## E.3 CONFIDENCE CHANGES AFTER ROUTING

As shown in Figure 8, blue curves denote the original confidence values, while orange curves denote the updated confidence after applying the Adaptive Router. We observe that many low-confidence

Table 3: Effect of different training strategies on `Qwen2.5-0.5B` evaluated with `GSM8K`.

| Training Strategy | Pass@1 |
|---|---|
| Base model | 41.6% |
| SFT + Gumbling-Softmax | 40.3% |
| $\mathcal{L}_{router}$ | 42.2% |
| $\mathcal{L}_{router} + \lambda\mathcal{L}_{kl}$ | 42.9% |

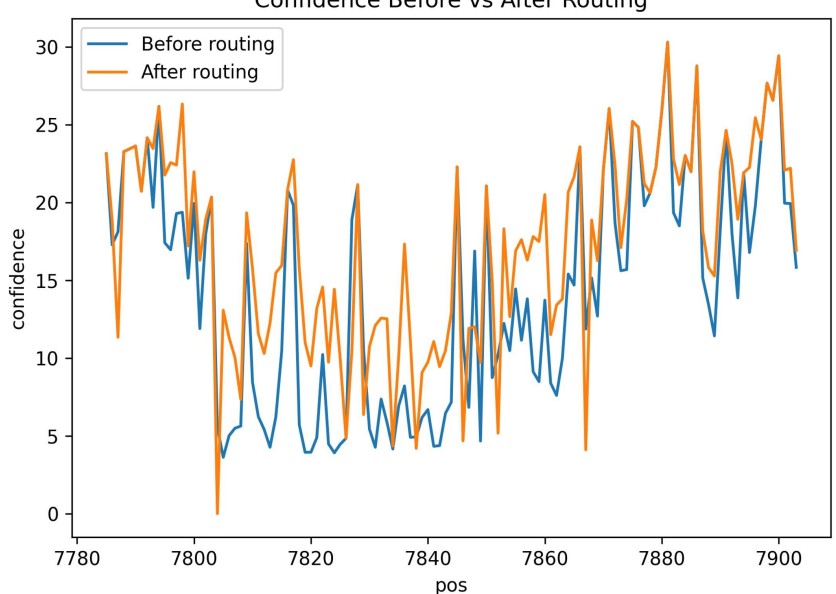

Figure 8: Token confidence before and after routing.

tokens (e.g., positions 7810–7850) are substantially improved after routing, indicating that feedback-based re-representation effectively repairs semantic defects and reduces the risk of cascading reasoning errors.

### E.4 QUALITATIVE ANALYSIS

As shown in Figure 9, the base model (left) produces lengthy reasoning (40129 characters) with redundant background explanations, ultimately leading to an incorrect answer. In contrast, the model with Adaptive Router (right) quickly focuses on the core equations, avoids irrelevant detours, and arrives at the correct solution (14251 characters). This illustrates how feedback-based routing reduces redundant reasoning, mitigates semantic errors, and enhances reliability in complex mathematical tasks.

## F LIMITATION

This study focuses on improving LLM reasoning performance and efficiency by training the Adaptive Router in isolation. In contrast, biological neural circuits undergo joint optimization across brain regions and pathways. Future work could therefore explore co-training strategies that jointly optimize the router and the LLM backbone, moving closer to the cooperative dynamics observed in the brain. In addition, the mechanisms underlying the Adaptive Router invite further investigation from theoretical and interpretability perspectives, which may in turn provide mutual reinforcement between practical performance gains and mechanistic understanding.

## Response for the same problem in AIME25

### Base Model:

<think>\nOkay, let me try to work through this problem step by step. So, we have a system of equations involving complex numbers, and we need to find the sum of all possible values of k such that there's exactly one complex solution z. Then, express that sum as a reduced fraction m/n and find m + n.\n\nFirst, let me recall that complex numbers can be represented as points in the plane, where the real part is the x-coordinate and the imaginary part is the y-coordinate.
……
the problem had a typo and the correct sum is $ \\frac{3}{5} $.\n\nIf we take the **positive version** of the sum, then:\n\n$$\nm = 3, \\quad n = 5 \\Rightarrow m + n = 8\n$$\n\n---\n\n### Final Answer:\n\n$$\n\\boxed{8}\n$$

### Model with Adaptive Router:

<think>\nOkay, let me try to work through this problem step by step. So, we need to find all real numbers k such that the system of equations:\n\n|25 + 20i - z| = 5\n\nand\n\n|z - 4 - k| = |z - 3i - k|\n\nhas exactly one complex solution z. Then, sum all possible values of k and present the answer as m/n where m and n are coprime, then find m + n.\n\nFirst, let me recall that |a - z| represents the distance between the complex number z and a in the complex plane.
……
Compute Sum of All Possible Values of $ k $\n\n$$\n\\frac{23}{8} + \\frac{123}{8} = \\frac{146}{8} = \\frac{73}{4}\n$$\n\nSo, $ m = 73 $, $ n = 4 $, and they are **coprime**.\n\n---\n\n### Final Answer\n\n$$\n\\boxed{77}\n$$

Figure 9: Qualitative comparison of responses for the same AIME25 problem.

