# OpenReview forum: "TAR: Token Adaptive Routing Framework for LLMs Token-level Semantic Correction Inspired by Neuro-Linguistic Pathways"
_ICLR.cc/2026/Conference — Submitted to ICLR 2026_

### Official Review · Reviewer_vYqw · 2025-10-30

**Soundness:** 2
**Presentation:** 2
**Contribution:** 2
**Rating:** 4
**Confidence:** 2

**Summary:**

Inspired by the structure of the human brain, this paper proposes TAR, a framework for token-level semantic correction. Based on the authors’ experiments, the proposed framework appears to yield certain performance improvements.

However, overall, I find the claimed connection between the method and biological brain structures somewhat tenuous. The paper does not provide sufficient justification or clear evidence for the claimed correspondence. This is my first time reviewing an AI paper that attempts to draw inspiration from biological structures, so I will lower my confidence accordingly.

**Strengths:**

**[S1]** The paper attempts to establish a connection between the proposed AI method and human brain structures, reflecting an interesting biological inspiration in the design of artificial intelligence systems.

**Weaknesses:**

**[W1]** The relationship between the proposed method and the biological brain structures needs to be strengthened. At present, the explanation feels rather superficial and unconvincing.

**[W2]** The paper devotes a large amount of space (around 7 pages) to describing the method and its connection to the brain, but the experimental and analytical sections are very limited (less than 2 pages). The paper feels more narrative-driven than methodologically innovative.

**[W3]** The experiments are insufficient. The evaluation is only conducted on small Qwen models, and several results are missing — e.g., GSM8K lacks Qwen3-1.7B evaluation, MATH500 lacks Qwen2.5-0.5B, and AIME25 lacks Qwen2.5-0.5B. Therefore, the results do not convincingly demonstrate the effectiveness of the proposed approach.

**[W4]** Token length is not an ideal metric for measuring efficiency, since the introduction of a router modifies the model structure. Reporting inference latency would provide a more reasonable and fair comparison.

**[W5]** The method description is not sufficiently clear. I spent a considerable amount of time trying to understand the proposed framework, and Figure 3 fails to clearly illustrate the core design.

**Questions:**

Please refer to the weaknesses section above.

---

> ### Author Response · Authors · 2025-12-01
>
> Thanks for your insightful feedback and comments. We will answer your questions in the following:
>
> ---
>
> **Q1: On the mapping between our method and biological structures**
>
> In the revised version, we will explicitly state that TAR is inspired by the **functional organization** of language-related pathways in the human brain (the ACC–IFOF–DLPFC/Broca loop), but does not attempt to establish a strict region-level or circuit-level mapping. We will de-emphasize “direct analogy” claims and instead focus on “functional analogy + modular inspiration.”
>
> ---
>
> **Q2: On balancing method description and experimental content**
>
> We strongly agree with your observation: the current version devotes more space to methods and neuroscience motivations, while the experimental and analytical sections in the main text are relatively compact. This is partly because token routing is a new module that requires more definitions and background. In the revision, we will:
>
> * Substantially shorten the early sections related to neuroscience;
> * Put more emphasis on experimental setup, results, and analyses, and, where appropriate, move key experiments from the appendix into the main text;
> * Achieve a better balance between “methodology” and “empirical evidence.”
>
> ---
>
> **Q3: On the choice of model–task combinations in the experiments**
>
> Our experimental design aims to demonstrate the effectiveness of TAR across models with different baseline abilities. One important detail is that router training relies on CoT traces generated by the target backbone itself, so the effectiveness of the router is inherently constrained by the backbone’s own capability. For Qwen2.5-0.5B, performance on MATH500 and AIME-level tasks is relatively weak, so we chose GSM8K—whose difficulty is better matched—as the main task for this model, to avoid a regime where the backbone fails on a large fraction of problems. In future work, we will further explore deploying the router on stronger backbones to improve the completeness of our results.
>
> ---
>
> **Q4: On complexity and efficiency evaluation**
>
> TAR uses a **sparse routing** design: our statistics show that only about 20% of low-confidence tokens trigger routing. The router itself consists of a single Transformer block, followed by a single forward pass through the target block, so the additional inference cost is controllable. We will include FLOPs and latency measurements in the revised version to quantify this more concretely.
>
> ---
>
> **Q5: On the readability of Figure 3**
>
> We will redesign and simplify Figure 3, including:
>
> * Clearly separating the training flow and inference flow in the diagram;
> * Reducing crossing arrows and unnecessary visual elements;
> * Using color and legends to more clearly distinguish modules and information flow.
>
> These changes should lower the cognitive load for readers and improve overall clarity.

---

### Official Review · Reviewer_5XsK · 2025-10-31

**Soundness:** 3
**Presentation:** 2
**Contribution:** 2
**Rating:** 4
**Confidence:** 3

**Summary:**

This paper proposes a token-adaptive routing framework to enhance the performance of LLMs by providing a second opportunity to answer questions at the token level. Inspired by neuro-linguistic pathways, the framework establishes a brain-inspired self-correcting loop that integrates seamlessly with LLMs without requiring additional fine-tuning. Through experiments conducted on two LLMs and comparisons with baseline methods, the framework demonstrates significant performance improvements.

**Strengths:**

1. The token-level framework does not require fine-tuning the LLMs, offering a self-correcting loop that enhances performance.

2. The framework design is grounded in theories of human brain function, making it more natural and theoretically sound.

3. The paper presents well-motivated research objectives, and the framework components effectively address these motivations.

**Weaknesses:**

1. As mentioned in the strengths, the framework does not fine-tune the LLMs directly, but instead tunes a router to enhance their representations. However, I would argue that this approach resembles LoRA-based fine-tuning, which also avoids modifying the main model parameters but introduces additional trainable components. Although the router operates differently from LoRA, both approaches still require training. From the paper, it is unclear whether the baseline model (i.e., Qwen) was fine-tuned with LoRA or not, but the router clearly involves training. Therefore, comparing a LoRA-based version and a router-based version would provide a fairer evaluation than the current setup.

2. The paper appears to draw inspiration from concepts in brain science, but the connections between these concepts and the proposed framework are not clearly established. Providing more details on how these ideas relate to the framework would help clarify the motivation and theoretical grounding.

3. There has been extensive research on enhancing LLM performance by modifying or reinterpreting their representations in a plug-and-play manner, without training any additional modules. How does the proposed framework compare in this regard? Can it function as a plug-and-play method, or does it necessarily require additional training?

**Questions:**

See Weakness

---

> ### Author Response · Authors · 2025-12-01
>
> We thank the reviewer for the valuable feedback and suggestions. We will answer your questions in the following:
>
> ---
>
> **Q1: On adding LoRA as a comparison baseline**
>
> As you rightly noted, introducing a router without modifying backbone parameters shares a high-level similarity with LoRA, and comparing the two is highly informative. Initially, we thought that since the training data are sampled from the model itself, LoRA fine-tuning might not dramatically change the knowledge boundary. However, we fully agree that for experimental rigor and completeness, LoRA should be explicitly included as a baseline. We will add the corresponding experiments in the revised version.
>
> ---
>
> **Q2: On the connection between neuroscience inspiration and the framework design**
>
> In the revised version, we will state more clearly that TAR is inspired by the **functional loop** of “ACC conflict monitoring – arcuate fasciculus/IFOF signal transmission – DLPFC/Broca error correction,” and does not claim a strict one-to-one biological correspondence. The key idea is to translate this “detect–route–re-represent” functional decomposition into the **which–where–how** three-stage structure in LLMs. We will streamline and rewrite the related sections so that the mapping between motivation and algorithmic modules becomes clearer.
>
> ---
>
> **Q3: On whether the router is plug-and-play (without training)**
>
> In Table 2 of Appendix E, we explored several rule-based random routing strategies that require no training. The results show that such non-learning-based routing does not provide stable performance gains and can even disrupt the original reasoning process. This indicates that the semantic functional distinctions inside a model need to be captured in a data-driven manner. Moreover, different LLMs—even different sizes within the same family—have different internal representation spaces and functional layer distributions. The router must learn the model-specific “defect–ability” mapping.
>
> Therefore, under the current framework, the router still needs to be trained for each specific model, and it is **not** fully plug-and-play.

---

### Official Review · Reviewer_4TYC · 2025-10-31

**Soundness:** 2
**Presentation:** 1
**Contribution:** 2
**Rating:** 2
**Confidence:** 3

**Summary:**

This paper introduces Token Adaptive Routing (TAR), a closed-loop mechanism that lets LLM detect and correct token-level semantic defects during inference without fine-tuning the backbone weights. The paper is motivated by how humans think and implement a "which-where-how" process. A semantic defect monitor to flag low-confidence tokens, an adaptive router that selects the most compatible model block, and lastly, a feedback-based representation that reinjects the token into the block to repair its semantics.

**Strengths:**

1. The papers introduces a TAR which is insipred by neuro-linguistic pathways that are present in the human brain
2. The method shows improved performance without fintuning the backbone weights.

**Weaknesses:**

1. The methodology section is hard to follow. The authors introduce a lot of components, where the motivation of each of the components within the router is missing. For example, the role of the ability vector and the requirement vector is hard to follow.
2. Does the method only select a single block? If yes, why is only a single block necessary for larger models, since multiple layers can be necessary to recalibrate a token
3. How many times does the self-correction loop go on? Is it a single loop, or is there an exit based on the confidence of the tokens?
4. Since the author introduces a lot of components in the training, ablation results are necessary to verify the importance of each component; however, most of the results have been moved to the appendix.
5. The authors train the model on AIME2024 and then test it on AIME2025; there is a strong chance of contamination here.
6. The model only focuses on math reasoning. Why was no experiment run on OOD tasks?
7. The paper chooses very small model,s and the router might not work properly when the number of layers increases in the model

**Questions:**

1. Do we require a separate router for each model?

---

> ### Author Response · Authors · 2025-12-01
>
> Thanks for your insightful comments and suggestions. We will address your concerns in the following answers:
>
> ---
>
> **Q1: On an intuitive explanation of the “ability” and “requirement” vectors**
>
> Our core idea is to view routing as a **semantic matching** problem:
>
> * The **requirement vector** describes what semantic abilities the current token is still lacking (analogous to a “lock”);
> * The **ability vector** characterizes which semantic abilities a given block is particularly good at providing (analogous to a “key”).
>
> The router’s role is to evaluate the lock–key compatibility and choose the most suitable block to complete and repair the token’s semantic representation. We will add this intuitive analogy in the main text to lower the conceptual barrier.
>
> ---
>
> **Q2: On the design choice between routing to a single block vs. multiple blocks**
>
> This is a very insightful question. In principle, routing a token through multiple blocks in sequence could further improve the repair effect. However, in practice, multi-block routing would multiply the computational and latency cost. In this work, our primary goal is to validate the effectiveness of the TAR framework itself, so we adopt the simplest configuration of **routing to a single target block**. In future extensions, we plan to include experiments with multi-block routing and compare performance–cost trade-offs at different routing depths.
>
> ---
>
> **Q3: On the number of iterations in the self-correction loop**
>
> In our current implementation, we perform at most one routing step, mainly because of the computational cost of large-model inference. We agree that multiple rounds of self-correction are a natural extension. In future work, we will add experiments with varying numbers of routing iterations and report their effects on both performance and inference latency, enabling a more systematic evaluation.
>
> ---
>
> **Q4: On the placement and clarity of ablation experiments**
>
> We have conducted ablations on the loss function design, the token routing strategy, and the router matching mechanism to verify the necessity and contribution of each component. These results are currently presented in the appendix, which may reduce their visibility. In the revised version, we will reorganize the paper structure and move the most essential ablation studies into the main text to make them easier for readers to access and interpret.
>
> ---
>
> **Q5: On potential data contamination between AIME24 training and AIME25 testing**
>
> We would like to clarify two points:
>
> 1. For AIME24, we only use the problem statements; the CoT traces used to train the router are generated by the target model itself and do not rely on external models or external training data, so no outside knowledge is injected.
> 2. AIME24 and AIME25 correspond to distinct competition years (2024 and 2025) with disjoint sets of problems, so there is no direct overlap or leakage between train and test.
>
> We will state this experimental setup more explicitly in the paper.
>
> ---
>
> **Q6: On the choice of OOD tasks in our experiments**
>
> We selected math tasks as our first testbed because they are highly formalized, have clearly verifiable answers, and place strict demands on local semantic consistency—making them an ideal “magnifying glass” for analyzing token-level semantic defects. Due to computational and space constraints, this paper focuses on GSM8K, MATH, and AIME. In future work, we plan to extend our evaluation to more typical OOD scenarios such as **linear logic tasks, graph-structured reasoning, and multi-hop QA**.
>
> ---
>
> **Q7: On the model sizes used in the experiments**
>
> We acknowledge that our current experiments are limited by available compute, and thus use smaller models like Qwen2.5-0.5B and Qwen3-1.7B. The patterns we observe (e.g., the relationship between confidence and error rate) are consistent across these models and align with intuition. When possible, we will add results on larger models (4B/8B) to further validate our conclusions.
>
> ---
>
> **Q8: On whether each model needs its own router**
>
> In our current setup, we train a **separate router for each backbone model**. The rationale is that different LLMs—even different sizes within the same family—exhibit markedly different internal representation spaces and layer-wise functional organization. The router must learn the model-specific mapping between “defects” and “abilities,” so it requires model-specific training.

---

### Official Review · Reviewer_JgFs · 2025-11-01

**Soundness:** 1
**Presentation:** 1
**Contribution:** 2
**Rating:** 2
**Confidence:** 4

**Summary:**

The paper presents TAR, Token Adaptive Routing, a method for self-correction of LLMs inspired by neuroscience, pathways in the brain that are non-directional and thus not only forward (one directional), for the task to correct incorrect LLM outputs.

The method is triggered when the output is of low confidence (threshold based), triggering an external adaptive router that looks at model-internal representations of layers ('ability vectors') to match best for so-called 'requirement vector' to match, to override the output layer ('to re-inject into block B_a() for feedback based re-presentation'). The adaptive router is trained with act as a router, by comparing SFT loss before and after routing to guide the router toward better policies.  In an LLM with L layers, each re-representation offers L + 1 routing options, and the paper tests all L + 1 policies to find the best. Finally, a regularization term is added to stabilize training.

The method is tested on three math reasoning benchmarks, GSM8K, MATH500 and AIME25. LLMs tested are Qwen2.5/0.5B and Qwen3/1.7B and trained on math data generated by itself. Each adapter is trained for one epoch.

**Strengths:**

- The paper presents a method for self-correction of an LLM in math domains.

- The paper is quiet well written (see comment below) and the method and setup is clearly explained.

**Weaknesses:**

- **Limited evaluation.** The evaluation is severely limited.

  - **Overly strong claim**. The paper motivates the method as " token-level semantic defects" detection method. However, and most importantly, the method is essentially a *self-correction method for math reasoning problems*, I strongly disagree with a claim for "semantic defects" when tested only on simple math problems. The title uses "linguistic pathways" which is too strong if tested in such a narrow domain where arguably linguistics is not really the key to solve the problem. Instead, the paper proposes a math error detection method. It would be stronger to compare and contrast to related work model-internal injections on natural language understanding (e.g. like the multi-hop natural language understanding problem in related work in Biran et al.). Moreover, the writing, especially in the introduction does not situate the method in math reasoning, but claims to provide a bigger human-inspired solution for "semantic defects", which I find misleading. There are no experiments beyond math problems, thus the claim of the current paper writeup is too strong and not supported by empirical validation.
  -  **Small LLMs only**. The method tests only two small LLMs and these are from the same model family (Qwen). This severely limits generalization. The method should be tested in at least two different model families to test generalizability beyond small Qwen models.
  - **Lack of comparison to upper bound or other method** The method does not compare against any existing method. For example,  the activation space like back-patching (Biran et al.'s method) could be applied by identifying a simple prompt that 'hints' at the solution by rephrasing the math problem (if the task is to solve 2 + 2 = 4, the test could be 1+3 = ?) and creating a probing classifier to identify the layer. This would also be a more lightweight approach and could help understand if the (quiet complex) method is useful. At least one comparison method should be included.
   - **Improvements** in Table 1 seem small (up to 3.2% accuracy). This raises again the question whether the routing method is useful.
   - **Lack of details of hyperparameter** The method relies on a confidence threshold (if the confidence if low, the routing fires. However, the paper does not provide any information of what threshold is used, nor how it was determined, nor how sensitive the method is to this threshold or how generalizable it is across the three math datasets. This is an important aspect left undermined.

- **Repetitive text parts**: The neuroscience inspiration part is quiet long and repeated 3x in the paper (abstract, into, method section)."TAR comprises three core components, each inspired by a function of the biological neuro-linguistic pathways ..."


- **Lack of complexity analysis** The method needs to identify which layer out of all layers for each token. This is very expensive. This is also perhaps why the paper only evaluates very small LLMs. The paper would be strengthened by providing a complexity analysis and judgement to what degree the method would scale up to larger LLMs.

**Questions:**

- how did you determine the threshold? how sensitive is your approach to the threshold?

---

> ### Author Response · Authors · 2025-12-01
>
> We appreciate the professional and insightful comments. We address each comment as follows:
>
> ---
>
> **Q1: On the relationship between our claims and the scope of evaluated tasks**
>
> Mathematical reasoning is a specific subtype of general language tasks. We chose math as our starting point because it imposes very strict requirements on semantic precision, making it a natural testbed for evaluating “semantic defect repair.” As discussed in the paper, cascading errors in math reasoning may be caused by token-level semantic deviations. We plan to extend our framework to broader tasks such as linear logic reasoning, graph search, and multi-hop question answering, so as to more fully support the overarching narrative of the paper.
>
> ---
>
> **Q2: On the choice of model sizes in our experiments**
>
> Our current experiments are indeed constrained by computational resources, which is why we focused on relatively small models such as Qwen2.5-0.5B and Qwen3-1.7B. Despite their modest size, we observe consistent and interpretable patterns across both models—for example, the relationship between token confidence and error rate. When resources permit, we plan to add experiments with larger models (4B/8B) to further validate the scalability of our approach.
>
> ---
>
> **Q3: On comparisons with existing methods (e.g., Back-patching)**
>
> We attempted to reproduce the method of Biran et al., whose main value lies in mechanistic analysis—showing that sending tokens back to earlier layers can improve multi-hop reasoning. However, at inference time, this method typically requires enumerating and evaluating a large number of routing paths, lacks an efficient inference pipeline, and results in high latency. Under our current setting, this makes it difficult to treat Back-patching as a practical online baseline.
>
> To provide a more informative comparison, we plan to include **LoRA fine-tuning** as a representative baseline in the revised version. LoRA represents a mainstream paradigm of “static parameter updates,” whereas TAR represents “dynamic computational routing.” Presenting them side by side will more clearly highlight the advantages and application scenarios of our method.
>
> ---
>
> **Q4: On interpreting the improvements reported in Table 1**
>
> We understand your concern regarding the magnitude of improvement. Here we would like to emphasize the difficulty and “high-score regime” nature of the AIME25 benchmark. AIME is derived from challenging competition-level math problems; on this dataset, even models that are several times larger in parameter count often differ only by a few percentage points. For example, within the same model family, Qwen3-8B outperforms Qwen3-4B by only about 1.7%. Against this backdrop, improving Qwen3-1.7B from 36.8% to 40.0% (+3.36%) **without modifying backbone parameters** is a practically meaningful gain. We will make this difficulty background and comparison more explicit in the paper.
>
> ---
>
> **Q5: On the choice of threshold and other hyperparameters**
>
> In Figure 4 of Appendix B, we analyze the relationship between token confidence and token-level SFT loss: when confidence > 20, the CE loss is almost zero, whereas in the region where confidence < 10, the loss increases sharply. Based on this empirical observation, we set the threshold to 20. In the revised version, we will move this justification into the main text and additionally discuss the sensitivity of our method to this threshold.
>
> ---
>
> **Q6: On the length and positioning of the neuroscience-inspired discussion**
>
> We agree that the neuroscience background section in the current version is somewhat lengthy. In the revision, we will:
>
> * Shorten the descriptions of biological mechanisms in the introduction and method sections;
> * Explicitly position them as **architectural inspiration**, rather than rigorous biological modeling;
> * Shift more space toward algorithmic details and empirical results.
>
> We believe these adjustments will improve both the focus and readability of the paper.
>
> ---
>
> **Q7: On the complexity analysis**
>
> Thank you for suggesting that we include a complexity analysis. By design, TAR adopts a **sparse routing** strategy: our statistics show that only about 20% of low-confidence tokens actually trigger routing. The router itself consists of a single Transformer block, followed by a single pass through the selected target block, so the additional inference cost introduced by routing is controllable. In the revised version, we will report FLOPs and latency measurements to quantify this more precisely.

---

### Author Response · Authors · 2025-12-01
**Summary for Area Chair**

We thank all the reviewers for their constructive comments and insightful suggestions. We have carefully added additional clarifications in accordance with the reviewers’ comments.

---

This paper proposes **Token Adaptive Routing (TAR)**, a dynamic self-correction mechanism that operates at the *token level* and improves reasoning without modifying backbone weights. TAR detects low-confidence tokens and selectively reroutes them to the **most semantically compatible layer**, forming a closed-loop correction process. Beyond empirical gains on challenging benchmarks such as AIME25, TAR offers insight into **reasoning in latent space** by turning token-level confidence into an actionable internal control signal. We observe consistent patterns between confidence, layer depth, and correction effectiveness, suggesting an emerging form of **token-level scaling behavior** inside LLMs. At present, research on **token-level routing for self-correction** remains relatively limited. TAR is simple, effective, and we hope it provides a valuable contribution to this underexplored direction in the community.

---

We are encouraged that the reviewers pointed out our work
``''self-correction in LLMs''``, ``''clearly explained''`` (**R JgFs**);
``''insipred by neuro-linguistic pathways''``, ``''improved performance without finetuning backbone weights''`` (**R 4TYC**);
``''a self-correcting loop''``, ``''grounded in human brain theories''`` and ``''addresses well-motivated research objectives''`` (**R 5XsK**);
``''interesting biological inspiration''``, ``''demonstrates performance improvements''`` (**R vYqw**).
We address the reviews below and will incorporate all changes in the revision.

---

### **1. Scope of the paper and generality of the claims**

We appreciate that reviewers found the core idea compelling and also encouraged us to better align the narrative with the empirical scope. We will refine our framing to emphasize that TAR is introduced and validated primarily in the context of **mathematical reasoning**, a domain that magnifies token-level semantic defects. At the same time, we will explicitly highlight how the design naturally extends to broader reasoning tasks, which we are actively exploring in ongoing work.

---

### **2. Experimental breadth and model scales**

Several reviewers suggested expanding the evaluation beyond small-scale Qwen models and math tasks. We agree that broader evidence will strengthen the paper. Mathematical datasets were chosen due to their precision and verifiability, and the consistent behaviors we observed across two different Qwen models give us confidence in the general applicability of TAR. We will clearly acknowledge resource constraints and outline plans for evaluating **larger models and more diverse tasks** in future work.

---

### **3. Comparison with existing baselines**

We appreciate the reviewers’ encouragement to include additional baselines. TAR and LoRA represent different forms of adaptation—dynamic vs. static—and we will add LoRA results for a clear comparison. While back-patching provides mechanistic insights, its inference-time overhead makes it difficult to serve as a practical baseline; we will clarify this distinction in the related work section.

---

### **4. Clarifying the neuroscience-inspired design**

Reviewers found the biological analogy interesting but suggested refining its presentation. We will condense the neuroscience discussion into a concise **architectural inspiration** section and focus on how the functional “which–where–how” loop translates into TAR’s algorithmic components.

---

### **5. Complexity, routing behavior, and efficiency**

We thank the reviewers for highlighting the importance of computational analysis. TAR employs **sparse routing**, with only ~20% of low-confidence tokens being rerouted. The router itself is lightweight, and the added cost is limited to a single forward pass through one block. We will provide **FLOPs and latency measurements** to quantify this overhead.

---

﻿Best regards,

The Authors

---

### Meta-Review · Area_Chair_d1xB · 2026-01-10

**Summary:**

This paper proposes a brain-inspired Token Adaptive Routing (TAR) framework to address cascading errors in LLMs' math reasoning, without fine-tuning the backbone model parameters. Experiments on small Qwen models (0.5B/1.7B) under math reasoning benchmarks show TAR improves accuracy and reduces inference tokens.

Main reviewer concerns: 1) Overly strong claims mismatched with the narrow math-only evaluation scope; 2) Limited evaluation on only small Qwen models; 3) Lack of comparisons with existing baselines; 4) Unclear methodological details (e.g., vector roles, confidence threshold setting, routing design rationale); 5) Superficial connection between TAR and neuroscientific mechanisms.

**Reviewer Concerns:**

Addressed Concerns:
- strong claims beyond empirical scope: Authors will refine framing to emphasize math reasoning context.
- insufficient comparison with baselines: Authors plan to add LoRA as a baseline for dynamic vs static adaptation comparison.
- Lack of complexity/efficiency analysis: Authors provide FLOPs and latency measurements to quantify overhead.


Outstanding Concerns:
- limited evaluation (small Qwen models only, narrow math tasks).
- Unclear neuroscience-method mapping

**Reviewer Scores:**

- Reviewer JgFs: likely to maintain reject (2) as core concerns on evaluation limitation and generalizability remain unaddressed.

- Reviewer 4TYC: may still keep reject (2) since methodology clarity and evaluation generality concerns are not fully resolved.

- Reviewer 5XsK: might increase scores as planned LoRA baseline addresses a major concern .

- Reviewer vYqw: might increase scores due to improved methodology presentation and planned efficiency analysis

---

### Decision · Program_Chairs · 2026-01-26

Reject